# A Novel Hierarchical Coding Progressive Transmission Method for WMSN Wildlife Images

**DOI:** 10.3390/s19040946

**Published:** 2019-02-23

**Authors:** Wenzhao Feng, Chunhe Hu, Yuan Wang, Junguo Zhang, Hao Yan

**Affiliations:** School of Technology, Beijing Forestry University, Beijing 100083, China; fengwenzhao36@bjfu.edu.cn (W.F.); huchunhe@bjfu.edu.cn (C.H.); wangyuan@bjfu.edu.cn (Y.W.); yanhao17@bjfu.edu.cn (H.Y.)

**Keywords:** progressive transmission, hierarchical coding strategy, wildlife monitoring image, saliency object detection, lossless and lossy coding

## Abstract

In the wild, wireless multimedia sensor network (WMSN) communication has limited bandwidth and the transmission of wildlife monitoring images always suffers signal interference, which is time-consuming, or sometimes even causes failure. Generally, only part of each wildlife image is valuable, therefore, if we could transmit the images according to the importance of the content, the above issues can be avoided. Inspired by the progressive transmission strategy, we propose a hierarchical coding progressive transmission method in this paper, which can transmit the saliency object region (i.e. the animal) and its background with different coding strategies and priorities. Specifically, we firstly construct a convolution neural network via the MobileNet model for the detection of the saliency object region and obtaining the mask on wildlife. Then, according to the importance of wavelet coefficients, set partitioned in hierarchical tree (SPIHT) lossless coding is utilized to transmit the saliency image which ensures the transmission accuracy of the wildlife region. After that, the background region left over is transmitted via the Embedded Zerotree Wavelets (EZW) lossy coding strategy, to improve the transmission efficiency. To verify the efficiency of our algorithm, a demonstration of the transmission of field-captured wildlife images is presented. Further, comparison of results with existing EZW and discrete cosine transform (DCT) algorithms shows that the proposed algorithm improves the peak signal to noise ratio (PSNR) and structural similarity index (SSIM) by 21.11%, 14.72% and 9.47%, 6.25%, respectively.

## 1. Introduction

Wildlife monitoring is crucial for the balance and stability of the whole ecosystem [1,2]. Images and videos of wildlife are the main materials that can be collected in the monitoring process. However, processing those materials in real time and effectively is a challenge. Conventional wildlife monitoring methods include crewed field survey, GPS (Global positioning system) collar [3], infrared camera [4] and satellite remote sensing monitoring [5] approaches. However, these methods have their own limitations, such as limited monitoring range, data acquisition lag and low resolution, and so on. Wireless multimedia sensor networks (WMSNs) [6] are applied in wildlife image collection as they present better deployment ability and accuracy.

Due to the limitations of WMSNs with low processing capability, power consumption restrictions and narrow transmission bandwidth, the wildlife monitoring images collected encompass high resolution and large information data characteristics, which poses a challenge for the WMSN transmission process [7]. For transmitting task through resource-constrained WMSN, image compression coding is utilized to reduce the transmission workload. In this field, image compression algorithms such as discrete cosine transform (DCT) [8], singular value decomposition (SVD) [9], and fast fourier transformation (FFT) [10,11] are capable of achieving high-efficiency compression of image samples. However, these algorithms are generally applied to the encoding and decoding processes of entire images, which means that the capability to transfer important regions in real-time is limited, besides, the transmission result is susceptible to external interference, such as transmission interruption and signal disturbance.

To solve the problem of unsuccessful transmission effectively, the image progressive transmission method [12,13,14] is utilized, in which the transmission process adopts the strategy that important regions are transmitted prior to other regions. However, the progressive transmission methods [15,16], which are based on the general displacement and staggered plane lifting method [17], cannot satisfy the practical demand for quick use of the reconstructed image information to distinguish the species of wildlife.

Saliency object detection can provide the mask of wildlife for the progressive transmission process, which is utilized to separate the object and background information. Conventional saliency detection algorithms, such as the human-computer interaction [18], visual attention model [19] and multilevel deep pyramid model [20] have undesired algorithm complexity issues.

This paper proposed a progressive transmission method for wildlife images based on saliency object detection. Firstly, regions containing saliency objects (wildlife) are detected from the original source by convolution neural networks, and then the mask of the saliency object region was generated. After the generation of the mask, set partitioned in hierarchical tree (SPIHT) [21] lossless coding transmission was performed preferentially on the saliency image to ensure the transmission accuracy of the wildlife region. Then Embedded Zerotree Wavelets (EZW) [22] lossy coding transmission was performed on the background region to improve the transmission efficiency, on the premise of ensuring the image reconstruction quality.

## 2. Wildlife Monitoring System

WMSN is widely used in wildlife monitoring systems to capture wildlife image materials with industrial grade cameras, which consist of WMSN terminal nodes, coordination nodes, gateway nodes and a data storage center (Back-end Sever). The wildlife monitoring system is acknowledged to present remote, real-time, all-weather and friendly monitoring merits, and the schematic diagram with detailed configurations are shown in Figure 1. Monitoring node devices developed by our laboratory are based on ZigBee network protocols and detailed parameters are shown in Table 1.

The WMSN monitoring system for wildlife monitoring was deployed in the Saihan Ula National Nature Reserve of Inner Mongolia Province, China. The experiment site has an average altitude of 1000 m above the sea level and it is temperate semi humid rainy climate. Wildlife species collected in the experimental site include *Cervus Elaphus, Lynx, Capreolus pygargus, Sus Scrofa* and *Naemorhedus goral*, and so on. *Cervus Elaphus* and *Lynx* are secondary national-protected species (shown in Figure 2). In this experiment, over 1600 images of more than 12 wildlife species were acquired and the total image data storage volume was 2.4 G.

## 3. Hierarchical Coding Progressive Transmission Method

A novel hierarchical coding progressive transmission method is proposed to process the wildlife monitoring images with high resolution and large information data, as shown in Figure 3. The region of interest (ROI) [23,24] that contains the wildlife is the major object of study, while background regions only provide comparatively irrelevant reference information. The steps of the algorithm are as follows:(1)Saliency region extraction based on convolution neural networks (CNN) [25], which are utilized to generate the mask image.(2)The maximum displacement method is applied to ensure the saliency region, in another words, the wildlife region is placed in the highest priority of compression transmission based on SPIHT coding.(3)To guarantee the transmission efficiency, the EZW coding algorithm is utilized to transmit the background region when the image information of the saliency region is received.

### 3.1. Mask Image Generation

Redundant background regions may cost undesired transmission bandwidth consumption, as background regions always account for a large percentage of the area of wildlife monitoring images. Therefore, extracting the wildlife region out of the background is the most significant procedure in our experiment. However, the high resolution and complex content in wildlife images aggravate the difficulty in completing saliency object detection and the mask image generation.

In our algorithm, the CNN is applied in the saliency object detection step, as it has been validated to detect the most visually distinctive object regions in the monitoring image. We selected 700 images as the training set, 100 images are used as the validation set and 200 images as the testing set, all of which are with 256 × 256 pixel size. The saliency object region is marked with pixels for model training, and the gray value 0/1 represents the background region/saliency object region.

Novel network architecture based on the MobileNet model [26] is utilized in this paper. In the conventional MobileNet model shown in Figure 4, the batch normalization (BN) method is dependent on batch size, BN’s error increases rapidly when the batch size becomes smaller. To solve the impact of batch size on the model, the group normalization method [27] is utilized in this paper, shown as

To obtain more detail and texture feature, skip layer short connections [28] which fuses the multi-level features extracted from different scales, is introduced into the network architecture, as shown in Figure 5.

In the network architecture, the side loss function and fusion loss function are calculated as
(1)$${\tilde{L}}_{side}(W,w,r)=\sum_{m=1}^{M}\alpha_{m}p_{side}^{\left(m\right)}(W,w^{\left(m\right)},r)$$
(2)$${\tilde{L}}_{fuse}(W,w,f,r)=\sigma (Y,\sum_{m=1}^{M}f_{m}{\tilde{R}}_{side}^{\left(m\right)})$$
(3)$$r=\left\{r_{i}^{m}\right\},i>m$$
where $\alpha_{m}$ is the weight of the mth side loss and $p_{side}^{\left(m\right)}$ denotes the cross-entropy loss function for side-outputs which is applied from [28]. σ(⋅,⋅) denotes the distance between the ground truth map and the fused predictions, and the corresponding continuous ground truth saliency map $Y=\left\{y_{j}^{\left(n\right)},j=1,\ldots ,\left|X_{n}\right|\right\},y_{j}^{\left(n\right)}\in \left\{0,1\right\}$.

Thus, we can get the final loss function
(4)$${\tilde{L}}_{final}(W,w,f,r)={\tilde{L}}_{fuse}(W,w,f,r)+{\tilde{L}}_{side}(W,w,r)$$

Hyper-parameters used in this section are learning rate (le-7), weight decay (0.0003), momentum (0.9) and loss weight for each side output (1). Our fusion layer weights are initialized at 0.2, and the group size is set to 32.

Regarding saliency object detection and extraction, the corresponding coefficient in the saliency and background regions are set to 1 and 0 respectively to obtain the binary mask image, as shown in Figure 6.

### 3.2. Progressive Transmission Strategy

After generating the mask of the wildlife region, we should determine which wavelet coefficients belong to the saliency region or the background region in the different levels of wavelet decomposition. In this paper, the mallat wavelet decomposition algorithm is utilized to obtain the wavelet coefficients, which are calculated by
(5)$$\left[\begin{array}{c} c_{j}[0]\\ c_{j}[1]\\ \vdots \\ c_{j}[\frac{n}{2}-1] \end{array}\right]=\left[\begin{array}{cccccccc} h[0] & h[1] & h[2] & h[3] & 0 & \cdots & 0 & 0\\ 0 & 0 & h[0] & h[1] & h[2] & h[3] & \cdots & 0\\ \vdots & \vdots & \vdots & \vdots & \ddots & \ddots & \ddots & \ddots \\ h[2] & h[3] & 0 & 0 & 0 & \cdots & h[0] & h[1] \end{array}\right]\left[\begin{array}{c} c_{j-1}[0]\\ c_{j-1}[1]\\ \vdots \\ c_{j-1}[n-1] \end{array}\right]$$
(6)$$\left[\begin{array}{c} d_{j}[0]\\ d_{j}[1]\\ \vdots \\ d_{j}[\frac{n}{2}-1] \end{array}\right]=\left[\begin{array}{cccccccc} g[0] & g[1] & g[2] & g[3] & 0 & \cdots & 0 & 0\\ 0 & 0 & g[0] & g[1] & g[2] & g[3] & \cdots & 0\\ \vdots & \vdots & \vdots & \vdots & \ddots & \ddots & \ddots & \ddots \\ g[2] & g[3] & 0 & 0 & 0 & \cdots & g[0] & g[1] \end{array}\right]\left[\begin{array}{c} c_{j-1}[0]\\ c_{j-1}[1]\\ \vdots \\ c_{j-1}[n-1] \end{array}\right]$$
where $h_{k}$ and $g_{k}$ are low/high pass filter respectively whose length we set is 4.

To ensure the integrity of the reconstructed image edge, the edge detection based on Canny operator is utilized in this paper, which aims to measure the convolution of the gaussian smoothing filter g and the above saliency detection result I(i,j) to obtain the most optimized approximation operator.
(7)S(i,j)=g∗I(i,j)
where S denotes convolutional result. ∗ refers to convolution function and (i,j) is the position of the pixel in saliency result.

Then the partial derivative is obtained by calculating the first-order finite difference of the filter result.
(8)P(i,j)≈(S(i,j+1)−S(i,j)+S(i+1,j+1)−S(i+1,j))/2
(9)Q(i,j)≈(S(i,j)−S(i+1,j)+S(i,j+1)−S(i+1,j+1))/2

Among them, P(i,j) represents the gradient partial derivative of image in x direction and Q(i,j) is the gradient partial derivative in y direction. Therefore, the pixel amplitude matrix and gradient direction matrix are calculated as shown in the Equations (10) and (11).
(10)$$M(i,j)=\sqrt{p{\left(i,j\right)}^{2}+Q{\left(i,j\right)}^{2}}$$
(11)θ(i,j)=arctan(Q(i,j)/P(i,j))

Finally, non-maximum suppression is completed through seeking amplitude maximum of the matrix along the gradient direction. The pixels with maximum amplitude are considered as the edge pixel. To make the image edge close, this paper selects double appropriate threshold (high threshold and low threshold). As a consequence, the non-edge points that do not satisfy the threshold condition are removed. Then the connected domain is expanded to get the final edge detection result.

In the process of wavelet decomposition, the mask image is updated simultaneously shown in Figure 7 to specify the classification of wavelet coefficients in the different levels.

For the preferential transmission of the important region, the maximum displacement transmission is applied in this section. The researchers are most interested in the wildlife target while the background region is a supplement for wildlife in the monitoring images. In this paper, we think highly of all wavelet coefficients in the saliency object region over the wavelet coefficients in the background regions. The background information is transmitted after the saliency object region information is completed, which helps to identify the wildlife species in the initial stage of transmission. The maximum displacement transmission strategy firstly calculated the bit plane layer s which according to the maximum value of the wavelet coefficients in background region.
(12)$$s=INT\left[\log_{2}\left|C_{max}\right|\right]$$
where INT denotes integer operation and $C_{max}$ represents the maximum wavelet coefficient in the background region.

When the value of the bit plane layer is determined, the wavelet coefficients in the saliency region are multiplied by $2^{s}$ to ensure that all wavelet coefficients are greater than the maximum value in the background region. At the decoding process, we only need to judge whether or not the wavelet coefficients are greater than $2^{s}$. If the coefficients are larger than $2^{s}$, we can determine that the coefficients belong to the saliency region. When the bit plane layer is reduced by $2^{s}$, the image wavelet coefficients can be restored to obtain the reconstructed image.

### 3.3. Saliency Object Region Transmission

Wildlife protection and wildlife scientific research, such as wildlife species identification and wildlife individual identification, both require high-resolution and high quality wildlife monitoring images, especially the wildlife itself. Therefore, after the bit plane layer of the saliency region is lifted, we utilized the SPIHT algorithm to achieve lossless compression transmission for saliency region coefficients, which guarantees the transmission quality of important region.

The spatial direction tree structure expressed in Figure 8 is utilized in this section to recombine image wavelet coefficients. In the spatial direction tree structure, each node is represented by coordinates, which are denoted as (i,j), and the coefficient of root node is denoted as c(i,j).

Where D(i,j) denotes the coordinate set of the descendant nodes that belong to node c(i,j), we can obtain the spatial direction tree by c(i,j)+D(i,j). The node O(i,j) represents the coordinate set of the direct descendant node that belongs to node c(i,j). L(i,j)=D(i,j)−O(i,j).

The function $S_{n}$ is proposed to evaluate the importance of coefficients, and the coefficient is important while $S_{n}=1$.
(13)$$S_{n}=\{\begin{array}{cc} 1, & max\left\{\left|c(i,j)\right|\right\}\geq T\\ 0, & otherwise \end{array}$$
(14)$$T_{0}=2^{\left\lfloor log_{2}(max\{\left|c(i,j)\right|\})\right\rfloor }$$
where *T* is the current threshold and the initial threshold is selected as $T_{0}$.

Considering the importance of wavelet coefficients according to $S_{n}$ in different frequency bands, the image wavelet coefficients are scanned by the order of zigzag, which is shown in the Figure 9. The coefficient with a large amplitude can be preferentially scanned, which preserves the main information and improve the quality of the reconstructed image.

In the wavelet coefficient scanning process, three lists are utilized to store the relevant information of the wavelet coefficients according to their importance: LIP (less significant pixels), LSP (significant pixels) and LIS (less significant sets).

The SPIHT rules set in this section include:(1)If the coordinate in c(i,j) is greater than the current threshold T, it will be stored in the LSP list.(2)All elements in D(i,j) are compared with the threshold T. The symbol D(i,j) is used to represent the coefficient set if there is no important element. Otherwise, the D(i,j) is divided into the O(i,j) set and L(i,j) set.(3)If there are important elements in O(i,j), they will be stored in the LSP.(4)The symbol L(i,j) is used to represent the coefficient set when there is no important element. Otherwise, the L(i,j) is splitted into four parts.(5)Step (5) is performed periodically for each newly generated spatial direction tree until all the important elements are stored in the LSP.

Finally, the wavelet coefficients are refined, coding by successive approximation quantization. The wavelet coefficients are continuously coded according to the update threshold $T_{i}=T/2$ until all wavelet coefficients in the saliency region are coded.

### 3.4. Background Transmission

To improve the transmission efficiency, the lossy compression EZW algorithm is implemented to encode background information, which has some similarities with the SPIHT coding algorithm.

In this section, the zero tree structure, whose definition is that the root nodes and all offspring nodes are unimportant coefficients, we can use an encoding symbol to represent all coefficients in a zero tree, thus greatly improving coding efficiency. In the EZW encoding process, the wavelet coefficients are divided into four symbols by comparing with the threshold T, as shown in Figure 10.

When the wavelet coefficient dividing process is finished, the wavelet coefficient scanning and refined coding is completed with the SPIHT coding algorithm in the last section until all coefficients in the background region are coded.

## 4. Comparison and Discussion

To verify the adaptability and effectiveness of the proposed algorithm, transmission analysis of field-captured wildlife monitoring images is presented. The result is evaluated by several evaluation criteria and it is compared to other conventional algorithms for image transmission.

### 4.1. Evaluation Criteria

Both peak signal to noise ratio (PSNR) and structural similarity index (SSIM) [29] are utilized as objective criteria to evaluate the quality of image reconstruction.

PSNR is the ratio of the signal maximum possible power to the destructive noise power based on the mean square error (MSE) [30], which affects representation accuracy.
(15)$$PSNR=10log\left[\frac{{\left(255\right)}^{2}}{MSE}\right]dB$$
(16)$$MSE=\frac{1}{MN}\sum {\left(g(x,y)-f(x,y)\right)}^{2}$$
where *MN* is the total number of pixels in the sample image. g(x,y) is the reconstruction image and f(x,y) is the original image.

The SSIM is another measure of the similarity between reconstructed and original images. It achieves this by calculating the image distortion degree according to the change of image structure information.
(17)$$SSIM(x,y)=\frac{\left(2u_{x}u_{y}+C_{1})(2\sigma_{xy}+C_{2}\right)}{\left(u_{x}{}^{2}+u_{y}{}^{2}+C_{1})(\sigma_{x}{}^{2}+\sigma_{y}{}^{2}+C_{2}\right)}$$
where $u_{x}$ and $u_{y}$ are the mean value of the luminance in the original and reconstruction image respectively. $\sigma_{x}$ and $\sigma_{y}$ are the standard deviation of the luminance. The constants $C_{1}$ and $C_{2}$ are used to suppress instability in structural similarity comparison.

### 4.2. Experiment Result and Analysis

We applied our algorithm to the filed-capture wildlife monitoring images with high resolution, high noise interference and complex background selected from own image dataset [31]. Experimental results are presented from left to right column in Figure 10 as a time progression by setting the bits per pixel (bpp) to 0.1, 0.4, 0.7, 1. All experiments were performed using MATLAB (2014a) in the workstation with Intel (R) Core (TM) i5-4570 and 4GB RAM.

As a proposed method, the saliency region is firstly transmitted through a lossless approach and then the background region in a lossy way [32], which are shown in Figure 11 by the first four and latter four columns separately. In the transmission process, we can quickly use the reconstructed image information to recognize the species of wildlife, such as the progressive transmission effect in columns 2–3. And we can also actively choose to terminate transmission after obtaining satisfactory information.

In the process of image transmission, the background region adopts lossy compression transmission to improve the efficiency of image transmission. We selected the PSNR and SSIM of the full reconstructed image to evaluate the proposed algorithm in this paper. The experimental results are shown in Table 2.

According to above transmission result in Table 2, the average value of PSNR and SSIM in the saliency region (the 4th column in Figure 11) and full image (the 8th column in Figure 11) are 46.1596 dB, 0.9876 and 39.0365 dB, 0.9014 respectively. Therefore, the proposed algorithm is capable of ensuring the reconstruction quality of the saliency object region and the full image reconstruction quality can meet the requirements of forest operators.

To verify the transmission effect, the EZW and DCT algorithms are compared with our proposed algorithm in this section. As shown in Figure 12, image quality is improved after adding the bpp. The reconstructed image quality corresponding to each bpp is improved when compared with EZW and DCT, which indicates that the scheme of preferentially transmitting the saliency region image is feasible. We can identify the wildlife species in the initial stage of transmission, which provides data support for subsequent research. Besides the average reconstructed results of PSNR and SSIM by our algorithm are 47.2515 and 0.9014 (bpp is set to 1.0), which increased by 21.11%, 14.72% and 9.47%, 6.25% respectively when compared with the EZW and DCT algorithms.

The average running time comparison result is shown in Table 3. The EZW algorithm cost the least calculation time, but the transmission quality is not ideal. Lossless coding can guarantee the quality of the transmitted image, while its running time cost is relatively high.

## 5. Conclusions

In this paper, we proposed a novel hierarchical coding progressive transmission method for wildlife images, which can achieve progressive transmission of saliency region and background separately. We firstly utilize a convolutional neural network based on MobileNet model to detect the saliency object region for a wildlife image with a complex background. When we obtain the mask image of the wildlife region, a progressive transmission strategy using the maximum displacement method aims to transfer the wavelet coefficients of the saliency region preferentially, and then lossless coding transmission is performed on the saliency region and lossy coding performed on the background region. To demonstrate the efficiency and validation of the proposed method, the images from the field-captured wild monitoring database are processed. Comparison results show that the proposed algorithm has better performance than existing classical algorithms, that is, EZW and DCT. Specifically, the average PSNR and SSIM are increased, respectively, by 21.11%, 14.72% and 9.47%, 6.25%.

## Figures and Tables

**Figure 1 sensors-19-00946-f001:**
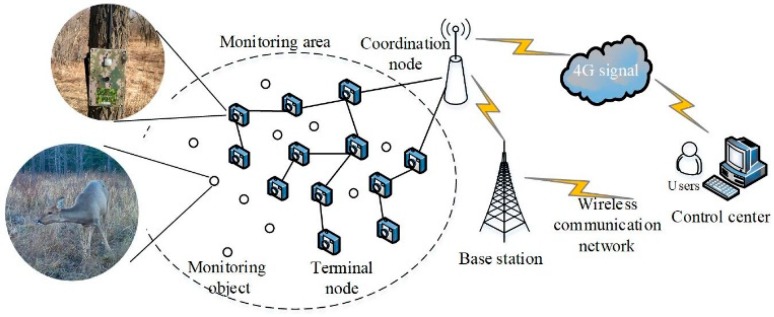
Wildlife monitoring system.

**Figure 2 sensors-19-00946-f002:**
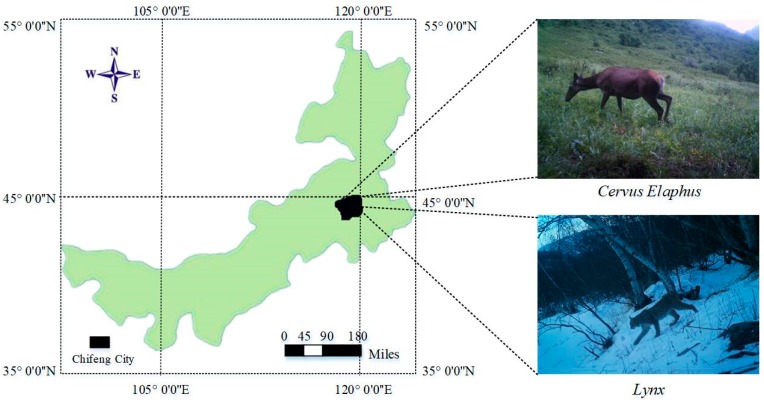
Wildlife monitoring images in Saihan Ula Nature Reserve.

**Figure 3 sensors-19-00946-f003:**
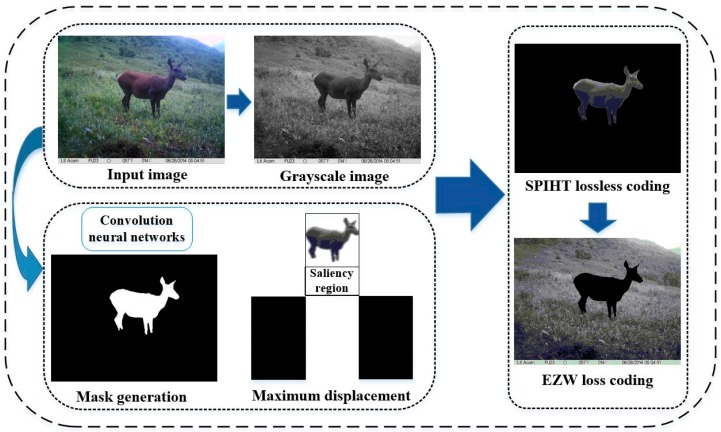
Structure of wildlife image progressive transmission process.

**Figure 4 sensors-19-00946-f004:**
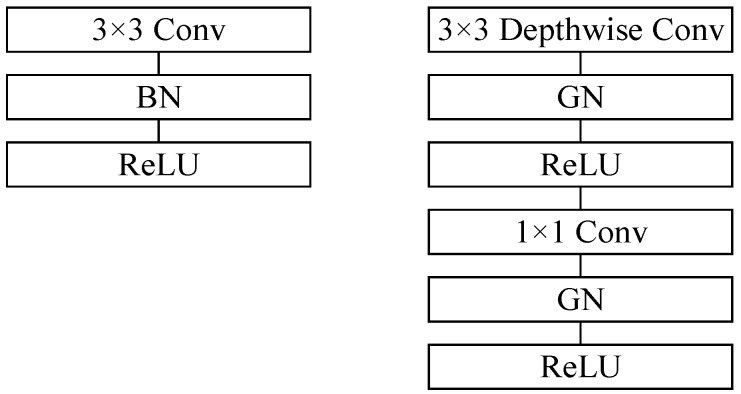
MobileNet Model proposed in our algorithm. **Left**: Standard convolutional layer with batch normalization and ReLU (Rectified Linear Units). **Right**: Proposed convolutional layer with group normalization and ReLU.

**Figure 5 sensors-19-00946-f005:**
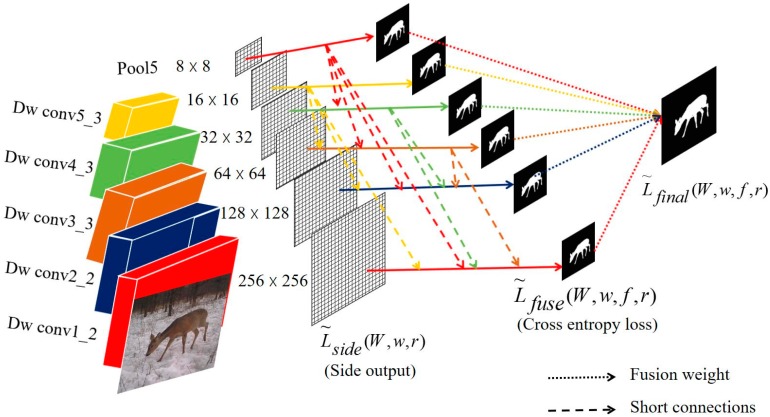
Skip layer short connections in network architecture.

**Figure 6 sensors-19-00946-f006:**
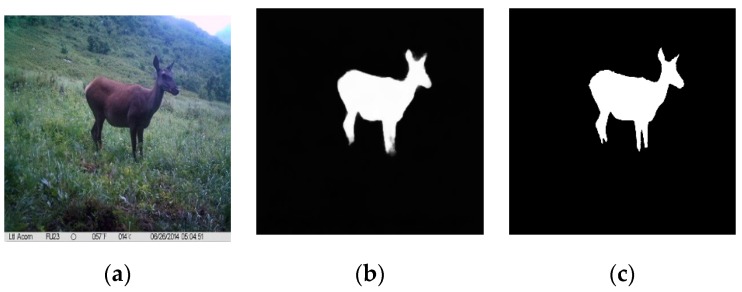
Saliency object detection and extraction. (**a**) Original image; (**b**) Saliency object region; (**c**) Ground truth.

**Figure 7 sensors-19-00946-f007:**
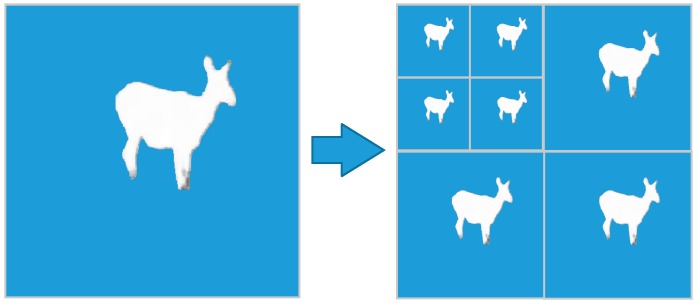
Mask update with wavelet decomposition. **Left**: The mask of saliency object region. **Right**: Mask update of three level wavelet decomposition.

**Figure 8 sensors-19-00946-f008:**
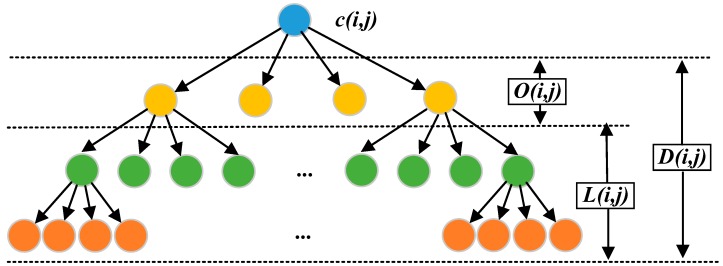
Three-level wavelet direction tree partition set.

**Figure 9 sensors-19-00946-f009:**
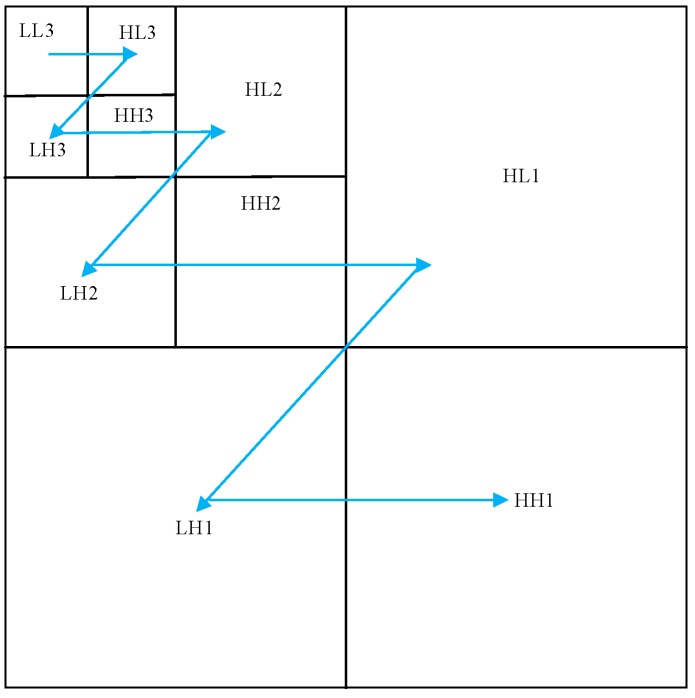
The coefficient scan order of zigzag.

**Figure 10 sensors-19-00946-f010:**
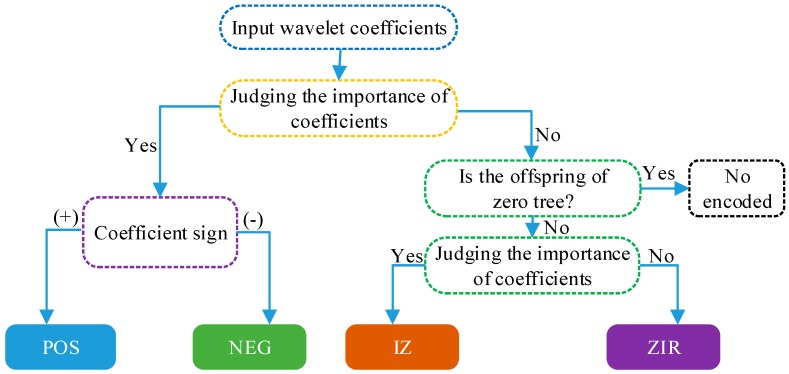
The encoding process of Embedded Zerotree Wavelets (EZW) algorithm.

**Figure 11 sensors-19-00946-f011:**
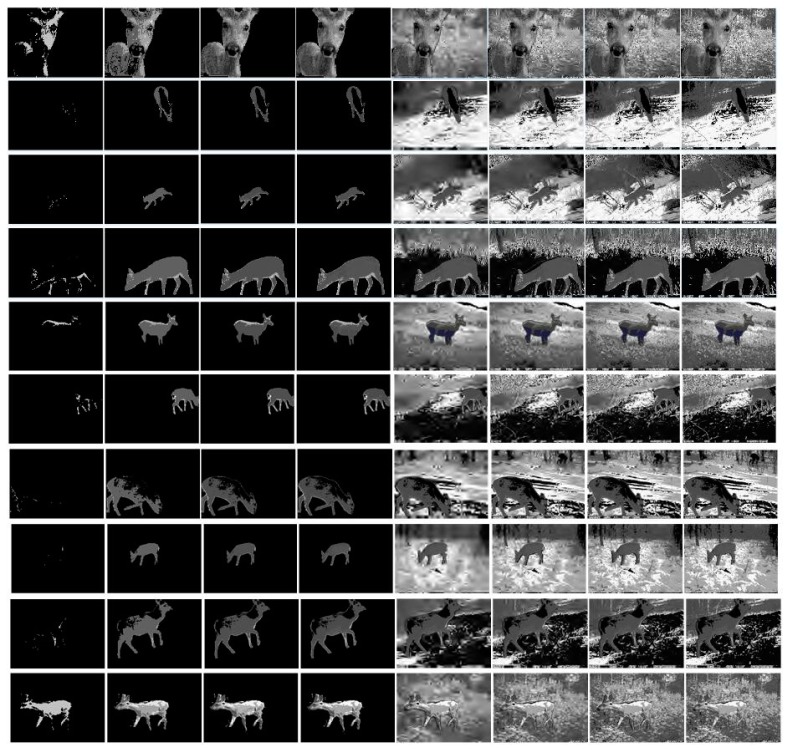
The flow chart of progressive transmission effect for wildlife images. The first 4 columns are the progressive transmission progress of saliency object region, and the last 4 columns are the progressive transmission progress of background region.

**Figure 12 sensors-19-00946-f012:**
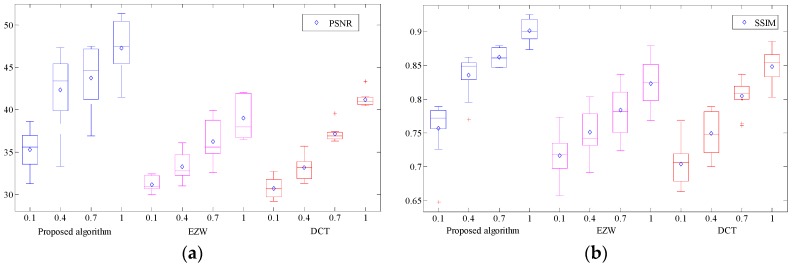
Comparison result of different algorithms. (**a**) Average peak signal to noise ratio (PSNR) experiment result. (**b**) Average structural similarity index (SSIM) experiment result.

**Table 1 sensors-19-00946-t001:** Parameters of wireless multimedia sensor network (WMSN) Node.

Monitoring Node	Function Parameters
Camera	OV7725 QVGA 30fps
Physical pixel	640 × 480
Support memory card	SD 16G
Controller	STM32 control ship (72 MHz CPU, 512 K SRAM)
Measuring range	310 m
Rate	200 kbps
Trigger mode	Infrared trigger

**Table 2 sensors-19-00946-t002:** The comparison effect of saliency region and full image reconstructed.

Sample Number	PSNR		SSIM	
	Saliency Region	Full Image	Saliency Region	Full Image
1	45.4356	41.4773	0.9772	0.8941
2	47.1674	39.2856	0.9894	0.9075
3	45.3462	37.5179	0.9779	0.8894
4	46.6589	41.1897	0.9897	0.9186
5	44.3027	36.9071	0.997	0.8872
6	44.5452	35.9571	0.9771	0.8734
7	45.4253	38.069	0.9967	0.8964
8	46.5923	39.6207	0.9977	0.9038
9	48.3964	42.2547	0.9856	0.9256
10	47.7263	38.0854	0.9881	0.9181

**Table 3 sensors-19-00946-t003:** Mean time comparison of transmission.

Method	Our Algorithm	Our Algorithm (Lossless Coding)	EZW	DCT
Average time(s)	4.132	17.631	3.974	9.379
Code type	Matlab	Matlab	Matlab	Matlab

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
