# Peer review of "A Novel Hierarchical Coding Progressive Transmission Method for WMSN Wildlife Images"

_sensors, 2019, doi:10.3390/s19040946_

Round 1

Reviewer 1 Report

The authors propose a hierarchical coding method for transmission of wildlife images.  The proposed method is based on CNN which allows to detect saliency map  that are further transmitted instead of all the images. The method is clearly explained. In addition, experimental results are provided and are convincing.

Minor correction: There is a typo in the title of Section 4.1.

Author Response

We thank the Reviewers and the Editor for their valuable suggestions toward improving our work as well as for their patience with this revision all along. The comments definitely helped us to substantially improve our work. In this letter, we provide responses to all questions raised by the Reviewers and we incorporate, as much as possible, in this new version of the manuscript all changes.

We have modified the manuscript based on the reviewers’ comments. The main changes made are highlighted in yellow and can be summarized as follows:

1. Expert comment: The authors propose a hierarchical coding method for transmission of wildlife images.  The proposed method is based on CNN which allows to detect saliency map  that are further transmitted instead of all the images. The method is clearly explained. In addition, experimental results are provided and are convincing.

Minor correction: There is a typo in the title of Section 4.1.

We have corrected the mentioned error in “ExperEvaluation criteria” of Section 4.1 and the corrected title is “Evalution criteria” in the revised manuscript. Besides we have fully proofread the whole manuscript and made all necessary revisions to ensure its quality. Those mistakes such as transferring, semi humid, splitted, are highlighted in yellow.

  Thank you again for your valuable comments! We sincerely look forward to your notice if there is any inadequacies during your review process.

Reviewer 2 Report

In this manuscript, the authors introduce a novel hierarchical coding progressive transmission method for wildlife images.

The paper is sufficiently well structured, and the proposed approach is described in a exhaustive manner (Section 3).

However, some parts of the paper should be improved. In particular, further details should be added for what concern the discussion of the achieved results (Section 4), especially in relation to the description of Figure 12 and Table 2.

As minor remarks, the authors should fix some typos, within the whole paper, as for instance, the following one:

- "MSE(Mean square error)..." -> "MSE (Mean square error)..." (line 215, page 8).

Finally, the quality of Figure 10 should be improved.

Author Response

We thank the Reviewers and the Editor for their valuable suggestions toward improving our work as well as for their patience with this revision all along. The comments definitely helped us to substantially improve our work. In this letter, we provide responses to all questions raised by the Reviewers and we incorporate, as much as possible, in this new version of the manuscript all changes.

We have modified the manuscript based on the reviewers’ comments. The main changes made are highlighted in yellow and can be summarized as follows:

1. Expert comment: In this manuscript, the authors introduce a novel hierarchical coding progressive transmission method for wildlife images.

The paper is sufficiently well structured, and the proposed approach is described in a exhaustive manner (Section 3). 

However, some parts of the paper should be improved.

1.1 In particular, further details should be added for what concern the discussion of the achieved results (Section 4), especially in relation to the description of Figure 12 and Table 2. 

The achieved results in section 4 have been further discussed and in detail figure 12 and table 2 are analyzed at length.

1) In the process of image transmission, the background region image adopts lossy compression transmission to improve the efficiency of image transmission. We need to calculate the PSNR and SSIM of the full reconstructed image which evaluate the proposed algorithm in this paper. The experimental results are shown in table 2. The experimental result show that the proposed algorithm is capable of ensuring the reconstruction quality of saliency object region and the full image reconstruction quality meet the requirements of forest operators. 

2) The Figure 12 shows that image quality is improved after adding the bpp and the reconstructed image quality corresponding to each bpp is both improved than EZW and DCT, which indicates that the scheme of preferentially transmitting the saliency region image is feasible. We can identify the wildlife species in the initial stage of transmission, which provide data support for subsequent research.

1.2 As minor remarks, the authors should fix some typos, within the whole paper, as for instance, the following one:

- "MSE(Mean square error)..." -> "MSE (Mean square error)..." (line 215, page 8).

We have corrected some spelling mistakes in the original manuscripts. For instance, MSE (Mean square error)line 215page 8was corrected intoMean squared error as suggested by reviewer. The whole manuscript was carefully proofread and thoroughly checked to eliminate potential mistakes. Those mistakes such as transferring, semi humid, splitted have been corrected and marked in the revised version.

1.3 Finally, the quality of Figure 10 should be improved.

We agreed to this comment and the figure 10 has been improved in this paper according the expert’s comments. The detailed revision is presented in table 10 of section 3.4.

  Thank you again for your valuable comments! We sincerely look forward to your notice if there is any inadequacies during your review process.

Reviewer 3 Report

The paper is poorly written, and before publication needs significant improvement. Detailed comments follows: 1) "transmission of wildlife monitoring images always suffers signal interference," Please explain why transmission in rural environment should suffer more from severe interference than transmission in urban area. ? 2) "To solve the problem of unsuccessful transmission process" Have you evaluated forward error correction (FEC) with data suitable interleaving. It really could solve the problem. 3) "SPIHT [21] lossless coding transmission was performed" SPIHT, especially lossless is very resource demanding algorithm, which means high power consumption, which poses a serious problem for "resource-constrained WMSN" Could you comment on it, please ? Is loss-less compression really necessary ? 4) "The maximum displacement method is applied ..." More information about this approach is necessary. Fig 3 does not provide enough explanation. 5) Equations (5) and (6) although represent image data filtering and downsampling but do not take into account border effect, which results in border distortion in the reconstructed image. 6) "the saliency region is firstly transmitted through lossless approach … which are shown in figure 11 by former 4 and latter..." Images in the first and second columns are definitely in lossy mode not lossless.

Author Response

We thank the Reviewer and the Editor for your valuable suggestions toward improving our work as well as for your patience with this revision all along. The comments definitely helped us to substantially improve our work. In this letter, we provide responses to all questions raised by the Reviewers and we incorporate, as much as possible, in this new version of the manuscript all changes.

We have modified the manuscript based on the reviewers’ comments, and the English language and style have been improved. The main changes made are highlighted in yellow and can be summarized as follows:

1. Expert comment: The paper is poorly written, and before publication needs significant improvement. Detailed comments follows:

1.1 "transmission of wildlife monitoring images always suffers signal interference," Please explain why transmission in rural environment should suffer more from severe interference than transmission in urban area. ? 

In the field, we used the wireless image sensor network based on ZigBee network protocol to monitor the wildlife. This system has weak signal transmission strength in the wild forest, which causes the transmission to be vulnerable to interference. Besides, due to the obstruction of trees and the fluctuation of mountain, the transmission signal will appear the reflection, refraction and diffraction in the wild environment which attenuates the transmission signal. Finally, the transmission in rural environment suffers more from severe interference than transmission in urban area is not proposed in this paper.

1.2 "To solve the problem of unsuccessful transmission process" Have you evaluated forward error correction (FEC) with data suitable interleaving. It really could solve the problem. 

Forward error correction(FEC) can improve the credibility of data communication. Once errors occur in transmission process, receivers are allowed to transmit by building data links again, while the end-to-end transmission reliability can not be guaranteed in multi-hop WMSN. Therefore the Hierarchical Coding Progressive Transmission Method for WMSN Wildlife Images was adopted in this paper. Saliency regions are firstly transmitted than background regions. Those saliency regions are transmitted in a lossless way while background regions are transmitted in a lossy way. The reconstruction quality of priority regions is guaranteed and the transmission efficiency of whole image is improved. 

1.3 "SPIHT [21] lossless coding transmission was performed" SPIHT, especially lossless is very resource demanding algorithm, which means high power consumption, which poses a serious problem for "resource-constrained WMSN" Could you comment on it, please ? Is loss-less compression really necessary ? 

The lossless compression indeed poses a problem for WMSN, which is capable of ensuring the reconstruction quality of images. However wildlife protection and wildlife scientific research, such as wildlife species identification, wildlife individual identification both require high-resolution and high quality wildlife montioring images, especially the wildlife itself. Our monitoring image samples acquired in field experiments mainly include Cervus Elaphus and Lynx, etc. Generally the target region always account for a small quantity in the whole image, therefore, this paper put more emphasizes on saliency target regions(wildlife regions). In this paper, lossless and lossy compression transmissions are conducted in saliency target regions(wildlife regions) and background regions respectively, which reduces the power demands of wireless sensor network and guaranteed the reconstruction quality of silency regions to meet practical needs of forestry researchers.

1.4 "The maximum displacement method is applied ..." More information about this approach is necessary. Fig 3 does not provide enough explanation.

In section 3.2, we added more information of maximum displacement method. According to actual application requirement, the researchers are most interested in wildlife itself, and the background region is a supplement for wildlife in the monitoring images. So we consider the importantance of all wavelet coefficients in saliency object region to be better than the wavelet coefficients in background region. The maximum displacement method firstly calculated the bit plane layer S which according to the maximum value of the wavelet coefficients in background region, and the wavelet coefficients in the saliency region are multiplied by 2^S to ensure that all wavelet coefficients are greater than the maximum value in the background region. The background information is transmitted after the saliency object region information is completed transmission, which we can identify the wildlife species in the initial stage of transmission.

1.5 Equations (5) and (6) although represent image data filtering and downsampling but do not take into account border effect, which results in border distortion in the reconstructed image.

We agreed to this comment and added edge detection method in the section 3.2 according the expert comments. To ensure the whole edges of reconstructed images, the edge detection method based on canny operator was utlized on the basis of image data filtering and downsampling in this paper. By computingthe pixel amplitude matrix and gradient direction matrix in imagesthe non-maximal value suppression is completed through seeking amplitude maximum of the matrix along the gradient direction. The pixels with maximum amplitude are considered as the edge pixel. Finally, to make the image edge close, this paper selects double appropriate threshold (high threshold and low threshold). As a consequence, the non-edge points that do not satisfy the threshold condition are removed.Then the connected domain is expanded to get the final edge detection result.

1.6 "the saliency region is firstly transmitted through lossless approach which are shown in figure 11 by former 4 and latter..." Images in the first and second columns are definitely in lossy mode not lossless.

In the figure 11, the first and second columns are the reconstructed images of saliency object region in transmission progress, not the final reconstructed images when the transmission completed. Each row in Figure 11 represents the progressive hierarchical transmission process of wildlife monitoring image. First 4 images are reconstructed images of saliency object regions whose bpp values are 0.10.40.7 and 1, while another 4 images are reconstructed images of background region in receiving server. To evaluate the effect of the proposed algorithm in this paper, the final evaluation of monitoring images are conducted by the fourth and eighth column images which are the final reconstructed images of saliency regions and full images.

  Thank you again for your valuable comments! We sincerely look forward to your notice if there is any inadequacies during your review process.

Round 2

Reviewer 2 Report

The authors have improved their work, according to the suggestions provided by the reviewer.